# Recombination of X-ray-Generated Radical Ion Pairs in Alkane Solution Assembles Optically Inaccessible Exciplexes from a Series of Perfluorinated *para*-Oligophenylenes with *N*,*N*-Dimethylaniline

**DOI:** 10.3390/ijms24087568

**Published:** 2023-04-20

**Authors:** Pavel V. Nikul’shin, Roman G. Fedunov, Leonid V. Kuibida, Alexander M. Maksimov, Evgeni M. Glebov, Dmitri V. Stass

**Affiliations:** 1A.V. Topchiev Institute of Petrochemical Synthesis, 119991 Moscow, Russia; npvru@mail.ru; 2V.V. Voevodsky Institute of Chemical Kinetics and Combustion, 630090 Novosibirsk, Russia; fedunov@kinetics.nsc.ru (R.G.F.); kuibida@kinetics.nsc.ru (L.V.K.); glebov@kinetics.nsc.ru (E.M.G.); 3N.N. Vorozhtsov Novosibirsk Institute of Organic Chemistry, 630090 Novosibirsk, Russia; maksimov@nioch.nsc.ru; 4Department of Natural Sciences, Novosibirsk State University, 630090 Novosibirsk, Russia; 5International Tomography Center, 630090 Novosibirsk, Russia

**Keywords:** exciplex, X-ray-generated luminescence, radical ion pair, perfluorinated oligophenylene, TDDFT, OLED, magnetic field effect

## Abstract

We demonstrate that a series of perfluorinated *para*-oligophenylenes C_6_F_5_-(C_6_F_4_)_n_-C_6_F_5_ (*n* = 1–3) produce exciplexes with *N*,*N*-dimethylaniline (DMA) in degassed X-irradiated *n*-dodecane solutions. The optical characterization of the compounds shows that their short fluorescence lifetimes (ca. 1.2 ns) and UV-Vis absorption spectra, overlapping with the spectrum of DMA with molar absorption coefficients of 2.7–4.6 × 10^4^ M^−1^cm^−1^, preclude the standard photochemical exciplex formation pathway via selective optical generation of the local excited state of the donor and its bulk quenching by the acceptor. However, under X-rays, the efficient assembly of such exciplexes proceeds via the recombination of radical ion pairs, which delivers the two partners close to each other and ensures a sufficient energy deposition. The exciplex emission is completely quenched by the equilibration of the solution with air, providing a lower bound of exciplex emission lifetime of ca. 200 ns. The recombination nature of the exciplexes is confirmed by the magnetic field sensitivity of the exciplex emission band inherited from the magnetic field sensitivity from the recombination of spin-correlated radical ion pairs. Exciplex formation in such systems is further supported by DFT calculations. These first exciplexes from fully fluorinated compounds show the largest known red shift of the exciplex emission from the local emission band, suggesting the potential of perfluoro compounds for optimizing optical emitters.

## 1. Introduction

X-irradiation with photon energies in the range of tens of keV is a convenient method of delivering energy to matter. In the case of organic materials with low stopping power, it can be used to create excited electronic states over large sample volumes almost uniformly; and for nonpolar solutions of luminophores, this is a useful method of exciting fluorescence irrespective of the absorption spectra and absolute absorbances of the samples, even for very optically dense solutions. The underlying mechanism of energy accommodation from keV down to eV level is rather complex and involves multistage radical ion chemistry, with the final step being the recombination of a radical ion pair, but it has been understood well enough to be able to controllably use it for inducing the excitation of target molecules. About a decade ago, it was discovered that it can also be used for the creation of excited-state complexes (exciplexes) between the parent molecules of the radical ion pair. It turned out that due to the close proximity of the partners of the recombining pair in the case of X-irradiation, exciplexes could be obtained for systems with arbitrarily short lifetimes of the local excited-state of the participating molecules. Together with the elimination of any restrictions on the optical absorption spectra of the components, this implied that such X-ray-generated recombination exciplexes could be experimentally produced for practically any suitable donor–acceptor pair that would energetically and sterically allow for this.

Starting with a well-known exciplex former, naphthalene (fluorescence lifetime τ = 96 ns), in pair with *N*,*N*-dimethylaniline (DMA) [1], X-ray-generated exciplexes were obtained for anthracene (τ = 5.9 ns) [2], *para*-terphenyl (τ = 0.95 ns) and diphenylacetylene (τ = 8 ps) [3], all paired with DMA. The diphenylacetylene–DMA exciplex, which could be generated only via the recombination channel, turned out to be the most convenient vehicle to create a number of related diphenylacetylene-based systems with systematically varied properties [4], develop methods to exploit the magnetic field sensitivity of the recombination [5] and estimate the exciplex lifetime [6]. A theory was developed to juxtapose the optical and X-ray generation of the exciplexes, augmenting the reaction scheme of radiation chemistry with the relevant processes leading to the experimentally observed exciplex formation [7]. The possibility to obtain unrestricted experimental data potentially opens vast opportunities for quantum chemical studies of exciplexes that require the most advanced methods to faithfully describe weakly bound, excited complexes of relatively large molecules. Furthermore, the radical ion pathway of radiation chemistry, mentioned above, closely resembles the electrochemical pathway operating in OLEDs; thus, X-ray-generated recombination exciplexes may be mimics or models of prospective emitters for organic optoelectronics [4]. In this work, we report a novel class of exciplex-forming systems, perfluorinated *para*-oligophenylenes paired with DMA.

Exciplex generation is an enormous and thoroughly studied field in photochemistry [8,9,10,11,12,13,14,15,16,17,18,19,20,21,22,23,24,25] and, to a lesser extent, in electrochemistry [26,27,28,29,30,31,32,33,34], with its established detailed reaction schemes and experimental protocols. Further important, more recent extensions here are the already mentioned use of exciplexes for optimizing emitters in OLEDs [35,36,37,38,39,40,41,42,43,44,45,46,47,48,49,50,51] and the quantitative rationalization of exciplexes in quantum chemical calculations [52,53,54,55,56,57,58,59,60,61,62,63,64,65,66,67]. Only several reports on radiation-generated exciplexes have appeared over the years [68,69,70,71,72,73], but they went virtually unnoticed. For the purpose of this work, it would probably be more useful to delineate what this paper is not about right away, to avoid misleading the reader into seemingly familiar territory with the unmet expectations and alert the experts about a different approach to things they already know so well. This work will focus on experimentally probing exciplex formation based on the study of newly arising emission bands in the spectra of X-ray-induced luminescence. Optical excitation will not be employed and is of no use for the systems of this work due to short excited-state lifetimes and overlapping absorption spectra with molar absorption coefficients so high, that the concentrations of components sufficient for exciplex generation via usual diffusion-controlled quenching of selectively excited donor by acceptor cannot be reached. This is the “optical inaccessibility” of exciplexes referred to in the title. Instead, the exciplexes will be assembled in situ from pairs of their component molecules, delivered close to each other while being the oppositely charged partners of radical ion pairs generated by X-irradiation of alkane solutions, and excited by back electron transfer at the moment of pair recombination. This is similar to the electrochemiluminescence scenario, but does not use charge injection from (micro)electrodes and support electrolytes to stabilize the radical ions. Instead, the pairs are constantly generated over the entire volume of a non-polar liquid sample by ionization events. Finally, this work will use magnetic field sensitivity of the exciplex emission band, which is a well-explored phenomenon in electrochemical [74,75,76] and photochemical [77,78,79,80,81,82,83,84,85,86,87] studies on more polar systems, where the exciplex is generated by reversibly quenching an excited molecule, and co-exists in equilibrium with a space-separated radical pair state. The radiation chemistry processes used in this work do not involve such a delicate balance of states in equilibrium and are irreversible, with the exciplex emission band simply inheriting the magnetic field sensitivity of the recombination of spin-correlated radical ion pairs in a low-polarity alkane liquid. The key elements of this approach in producing and studying exciplexes have been described earlier, in a series of our publications [1,2,3,4,5,6,7], to which the reader is referred regarding details and a historical overview. In this work, we shall focus on a specific set of new, promising exciplex-forming systems, quoting necessary details only when they are required for describing a particular experiment as the story unfolds. The experiment is supported by DFT calculations that reproduce exciplex formation in such systems.

The paper is organized as follows. First, the relevant optical properties of a series of perfluorinated *para*-oligophenylenes are determined. A section with a brief description of the specifics of luminescence studies in radiation chemistry follows, that explain the experimental choices of this work. Then, in three sections, we experimentally demonstrate the exciplex formation as revealed by a new emission band in the X-ray-induced luminescence spectra, isolate the exciplex spectra by selective quenching of the longer-lived exciplex emission with dissolved oxygen, and verify the recombination pathway of exciplex assembly by demonstrating the magnetic field sensitivity of the exciplex emission band. Three sections on DFT calculations then demonstrate that bound excited states are indeed possible in such systems, providing reasonable structures of radical ions and excited states, and spectral properties of perfluorinated biphenyl and *para*-terphenyl, predicting the position of exciplex emission bands and explaining the experimentally observed qualitative differences between perfluorinated biphenyl and its larger homologues. The Section 3 provides the necessary details on the synthesis of the compounds, solvent preparation and experimental setup for taking the X-ray-induced luminescence spectra. Full sets of optical characterization data and tables from the DFT calculation are collated in the Appendix A. Unless specifically referenced, all optical data on the known molecules are taken from [88].

## 2. Results and Discussion

### 2.1. Optical Characterization of Compounds

Figure 1 shows UV-Vis absorption spectra of decafluorobiphenyl **1** (Figure 1a), perfluoro-*para*-terphenyl **2** (Figure 1b), perfluoro-*para*-quaterphenyl **3** (Figure 1c) and perfluoro-*para*-quinquephenyl **4** (Figure 1d) in acetonitrile, and Figure 2 shows their emission spectra in acetonitrile. Fluorescence kinetics and excitation spectra **1**–**4** are presented in the Appendix A, which also includes the set of optical characterization data in the second solvent, *n*-dodecane. The results for the two solvents are very similar, with no noticeable solvatochromism. However, the spectra/kinetics for *n*-dodecane are less straightforward to process due to artifacts from the unconventional solvent for fluorescence measurements. Table 1 summarizes the optical characteristics of the four compounds in acetonitrile, listing positions of absorption maxima and molar absorption coefficients at the maxima, positions of emission maxima, fluorescence quantum yields and lifetimes.

It can be seen that the properties of compounds **2–4** are rather similar, demonstrating a systematic and reasonable variation with an increase in the number of phenylenes, while perfluorobiphenyl **1** stands out in this series and its similarity with higher oligophenylenes is superficial. This distinction between **1** and higher oligophenylenes **2–4** will be often met throughout this work and is probably related to the more relaxed structures of excited states and radical ions of the higher oligophenylenes due to the presence of “inner” phenylene moieties. The single absorption maxima of **2–4** monotonically shifts to longer wavelengths, from 245 to 257 nm, with an increase in the number of phenylenes, the molar absorption coefficients monotonically increase from 2.7 × 10^4^ M^−1^cm^−1^ to 4.6 × 10^4^ M^−1^cm^−1^, emission maxima monotonically shift to the red from 324 to 332 nm, and the fluorescence quantum yields monotonically decrease from 4.3% to 3.5%, which is accompanied by a slight monotonic shortening of the fluorescence lifetime from 1.23 to 1.17 ns. On the other hand, decafluorobiphenyl **1** shows a double-peak absorption at 222/267 nm, with significantly lower molar absorption coefficients, an emission band with a very large Stokes shift, and a nearly halved quantum yield and doubled lifetime of fluorescence.

### 2.2. Settings, Samples and Spectra of X-ray-Induced Luminescence

While this work operates with notions and includes certain experiments typical for photochemistry, at the core, it is a radiation chemistry study and critically depends on the idiosyncrasies of fluorescence generation under ionizing radiation. Therefore, some choices of experimental options, such as solvents, concentrations, ampoule material, etc., may at first seem unusual or suboptimal for experts in photochemistry, who expect to see familiar protocols. Even the use of nomenclature may seem counterintuitive from certain aspects. Thus, while in the context of photochemically generated exciplexes it is customary to refer to “donor” and “acceptor” as “energy donor,” i.e., the molecule that is selectively excited into its characteristic absorption band and will serve as the source of energy for exciplex formation, and “energy acceptor,” i.e., the molecule that quenches the excited donor to eventually form the exciplex, in radiation chemistry, these terms normally apply to electron donors and acceptors that are used to produce the radical cation and radical anion, respectively, of the target compounds upon charge capture after solvent ionization. In the context of this work, the oligophenylenes would have been “energy donors” had it been possible to excite them selectively in the presence of the required amounts of the second partner, and DMA would have been an “energy acceptor.” From the radiation chemistry viewpoint, DMA is an electron donor and forms radical cation via capturing the solvent hole, and the perfluorinated oligophenylenes are electron acceptors that can only capture free electrons formed upon solvent ionization to produce radical anions. Both the electron donor and acceptor are also simply referred to as charge acceptors. Therefore, this section is included to briefly explain why and how the radiation chemistry experiments were performed, and why the spectra look as they do. A more detailed description of the relevant radiation chemistry can be found in more specialized publications [89,90,91,92].

Irradiation of saturated hydrocarbons by X-rays with quanta energies tens of keV, as used in this work, results in the ionization of solvent molecules to produce the primary radical ion pairs of solvent radical cations (hole) and free electrons, with a characteristic initial separation of ca. 10 nm. This is well within the Onsager radius for the nonpolar liquid (ε ca. 2), and the two oppositely charged species are mutually attracted by Coulomb interaction and eventually recombine with close to 100% probability, forming a solvent-excited state upon recombination due to the energy released upon back electron transfer. If suitable electron donors (Ds)/acceptors (As) are added into the solution, the primary radical ions can be captured to form secondary radical ions, D^+•^ and A^−•^. Due to the high electron mobility, useable concentrations of A are rather low, starting from ca. 10^−4^ M and going up to several units of 10^−3^ M. The probability of the capture increases with increasing concentration in A sublinearly, as [A]^0.7^, due to the peculiarities of electron mobility mechanism [93,94,95]. A pair with a captured electron consists of partners with molecular mobility and has typical recombination times of the order of tens of ns, and the competing capture of the radical cation in a diffusion-controlled reaction requires higher concentrations of the donor, in the range of 10^−2^ M. A recombination of any pair comprising D^+•^ and/or A^−•^ produces an electronically excited state of the additive and, if it is a luminophore, may produce a quantum of luminescence. C–H polyaromatics, such as naphthalene or biphenyl, can capture both electrons and holes in alkanes. However, aromatic amines in alkanes do not capture electrons and efficiently capture holes, and polyfluorinated compounds do not capture holes and are excellent electron acceptors. We shall use DMA for D and perfluorinated *para*-oligophenylenes for A, and focus on the recombination of the D^+•^/A^−•^ radical ion pair that produces the two molecules, D and A, at the distance of back electron transfer, which, in this case, is about 1 nm or less. Additionally, the back electron transfer is accompanied by an energy deposition sufficient to produce both local excited states of the acceptor luminophores, which is the commonly accepted paradigm, and exciplexes, which we shall explore in this work.

The described picture has several far-reaching implications that determine the acceptable experimental choices. A nonpolar solvent with sufficiently stable radical cations is required, which excludes options such as water, alcohols, acetone, or acetonitrile. Chlorinated solvents capture electrons dissociatively and are unsuitable. As the observed signal is luminescence and we plan to study the emission spectrum from the sample, the solvent should show minimal intrinsic luminescence, which excludes aromatic liquids such as benzene and its methylated derivatives. The most convenient remaining option is the quoted saturated hydrocarbons, alkanes. Of the available varieties of alkanes, cyclic alkanes, such as cyclohexane [96,97,98,99] and decalins [100,101], have unusually high mobilities of the solvent hole, which is a disadvantage in this case. Branched alkanes, such as neopentane and 2,2,4-trimethylpentane, have radical cations with controversial properties also best avoided [102,103,104], narrowing the choice to *n*-alkanes. Finally, a reasonably high viscosity and boiling point are preferable to allow for the sufficient time for pair recombination and the improvement of the sample stability in evacuated ampoules (see below), which, in practice, excludes the lighter liquid alkanes such as pentane or hexane. Thus, it can be seen that all the “usual suspects” for spectrophotometry studies that can be obtained at the HPLC/spectrophotometry purity grade have been excluded. Of the readily available *n*-alkanes, the best overall compromise has been found in *n*-dodecane (bp. 214.5 °C) used in this study. However, this solvent has never been considered suitable for spectroscopy, and is thus only available at a reagent grade level of purity. It is produced commercially by distillation from natural oils, and contains significant amounts of unsaturated/aromatic impurities, completely overshadowing the 10^−2^–10^−4^ M levels of additives that are introduced deliberately. Therefore, the solvent has to pass a deep purification procedure, as outlined in Section 3, which lowers the content of impurities to ppm levels. Nevertheless, this is still unsuitable for the spectroscopic characterization of compounds, which was done in acetonitrile here, but is acceptable for the radiation chemistry experiments described below.

The sample is a liquid and must be contained in some sort of ampoule. The underlying radiation chemistry involves radical ions and, as we shall see, long-lived excited states, thus the samples should be oxygen-free. Bubbling with Ar or N_2_ is not sufficient, and the usual option is the freeze–pump–thaw cycling at a forepump vacuum post. Thus, the ampoule must withstand thermal cycling to liquid nitrogen temperatures, which practically means round tubes with either a Teflon stopcock for reusable ampoule, or a removable stopcock adapter with sealing for one-off samples, as used in this work. The usual best material for optical-quality ampoules, quartz/suprasil, is unusable for spectrally resolved luminescence studies under X-rays due to the rather intense intrinsic luminescence covering the needed wavelength range. Irradiation of quartz produces stable defects commonly referred to as F-centers, where “F” stands for “fluorescence,” and such ampoules show an intense and constantly growing in time fluorescence in the range 250–500 nm [105]. The second-best choice for ampoule material, pyrex, which can be readily obtained in the form of calibrated thin-walled NMR tubes, also produces intense emission. The best results have been obtained earlier with tubes made from molybdenum glass, the material used in this work. For further details regarding the choice of ampoule material see ref. [1]. Sample preparation is described in Section 3. However, as any glass, it has a short wavelength absorption cut-off at about 310 nm, and therefore all X-ray-induced luminescence spectra presented in this work start at ca. 310 nm. Furthermore, the sample contains rather high amounts (ca. 10^−2^ M) of charge acceptors with molar absorption coefficients of up to several tens of thousands M^−1^cm^−1^, which leads to a significant inner filtering effect. Therefore, the shorter wavelength band at about 340 nm is heavily distorted by inner filtering and a short wavelength cut-off, and is not an emission band in the spectroscopic sense. This wavelength band is present in all spectra and comes from the overlapping local emissions of oligophenylenes and DMA, excited either by the recombination of pairs including one or both of these molecules, or by harvesting excitation from solvent molecules. However, the red-shifted exciplex emission band that arises in these systems is spectroscopically pure, as this is the red-most emission in the system in the wavelength range where there is no absorption from the solution, nor the ampoule. It is also an advantageous coincidence that it does not yet overlap with the second-order diffraction phantoms from the 340 nm band in the region of ca. 700 nm, and already does not overlap with the residual emissions from the glass centered at ca. 400 nm [1].

The final factor that needs to be discussed is the unusually high concentrations of compounds in the context of photochemical studies. While the spectral consequences of this have already been mentioned, with rod-shaped molecules such as perfluorinated *para*-oligophenylenes, a separate problem that arises is their limited solubility in the acceptable solvents. First, starting with the optical absorption spectra in acetonitrile of the previous section, they were obtained as follows. A 3 × 10^−3^ M solution of the compound in acetonitrile was prepared, and then added in 10 μL quantities to 3 mL of acetonitrile in a 1 cm optical cuvette, to obtain the UV-Vis spectra, each step corresponding to a 1 × 10^−5^ M increase in concentration. Finally, the most suitable spectrum of the series was chosen. This procedure worked for **1** and **2**, but for larger oligophenylenes, the concentration of the mother solution had to be decreased down to 1 × 10^−3^ M for **3** and 5 × 10^−4^ M for **4**, which was within a factor of 2 below their solubility limits in acetonitrile. For *n*-dodecane, the solubility was expectedly lower still. The target concentrations of the charge acceptors were standardly set at 5 × 10^−3^ M for A (perfluorinated *para*-oligophenylenes) and 1 × 10^−2^ M for D (DMA), and the samples were prepared by preparing mother solutions with twice these concentrations and then diluting the solution of A 1:1 either with neat *n*-dodecane (samples containing only the compound) or with the 2 × 10^−2^ M solution of DMA. This procedure worked only for **1**; for the larger oligophenylenes, the target amounts could not be dissolved. The mixtures were sonicated for 1 h, left for settling down, and then the obtained saturated solution from the supernatant was used for sample preparation. The concentrations were determined by adding a 30 μL bolus of the saturated *n*-dodecane solution into 3 mL of acetonitrile in a 1 cm optical cuvette, vigorously shaking to extract the contained compound (*n*-dodecane and acetonitrile are immiscible), measuring the absorbance, converting it to the concentration in acetonitrile and then converting it to the concentration in the parent *n*-dodecane solution. In this way, the concentrations that are indicated in captions to Figs. 3–5 were obtained: 5 × 10^−3^ M for **1**, 1.8 × 10^−3^ M for **2**, 4.4 × 10^−4^ M for **3**, 1.1 × 10^−4^ M for **4**. However, even these should be considered as upper estimates, as in the process of freezing/thawing during sample degassing and sealing the precipitate formed, that was not completely dissolved for **3** and **4**. While the specific concentrations are not critical for the reported study, it should be noted that the concentrations of the larger oligophenylenes grew progressively lower, and for **4**, reached the practical limit for radiation chemical experiments. This, together with diminishing synthetic yields (see Section 3), sets the practical limit for the range of the studied systems.

### 2.3. Exciplex Formation

Figure 3 shows the spectra of X-ray-induced luminescence for the four compounds and their mixtures with DMA in degassed *n*-dodecane solutions. Each panel also includes the spectrum from the sample containing only 10^−2^ M DMA. It can be seen that for a sample containing only one component, either oligophenylene **1**–**4**, or DMA, there is only the short wavelength emission band at ca. 340 nm, the distorted band of local emission of **1**–**4** or DMA (which is also a luminophore with φ = 0.11 and intrinsic emission band at 320–370 nm). However, when both components are present in the solution simultaneously, in the spectra of **2**–**4**, a new red-shifted band arises in the region of 500 nm that is attributed to the exciplex between the polufluorinated *para*-oligophenylene and DMA. No new exciplex emission band appears for decafluorobiphenyl **1**, which, again, sets it apart from the larger oligophenylenes **2**–**4**. Similar X-ray-generated exciplexes have been reported earlier for naphthalene/DMA [1,6], anthracene/DMA [2,6], *para*-terphenyl/DMA [3,6], diphenylacetylene/DMA [3,6], and several trifluoromethyl- [4,6] and methoxy-substituted diphenylacetylenes with DMA [4]. These earlier works reported a detailed study of exciplex formation for varying concentrations of the components and included comparison with optical excitation in those systems that supported this, with the conclusion that the standard “optical” pathway of exciplex formation in the conditions of the reported experiment is possible only for luminophores with fluorescence lifetime in excess of several nanoseconds (with suitably high concentrations of DMA); while for *para*-terphenyl (τ = 0.95 ns) and luminophores with still shorter lifetimes of fluorescence, the only operable channel is via the recombination of a radical ion pair. An “optical” pathway via the excitation of DMA (τ = 2.4 ns) and its quenching by the other partner is not effective at the used concentrations of the electron acceptor that, in this case, should perform as the quencher.

It is interesting to note that the reported exciplexes are the first X-ray-generated exciplexes, and to the best of our knowledge, the first exciplexes of any type produced from fully fluorinated electron acceptor/“energy donor,” and they demonstrate the most red-shifted emission of the previously reported systems and the largest separation between the local emission and exciplex emission bands. Whether this is a general feature of perfluorinated systems is an open question worth exploring, as this enhances the possibilities of tuning the exciplex emission band both in terms of the target color of emission and in terms of improving the light recovery by moving the emission band farther away from any absorptions in the system. It can further be noted that the exciplex emission band tends to shift to red with increasing the number in phenylenes within the molecule.

### 2.4. Quenching of the Exciplex Emission Band

The spectra of Figure 3 contain two emission bands, the dominant shorter wavelength one in the region of 340 nm, which, as described above, is a heavily distorted band of local emission from the two components of the mixture, and the longer wavelength exciplex band in the region of 500 nm that is the essence of this work. It is possible to experimentally isolate the exciplex emission band to a certain extent by taking advantage of the significant difference in emission lifetimes of the two types of emitting species following the procedure described in detail in ref. [6]. Both the perfluorooligophenylenes **1**–**4** and DMA have fluorescence lifetimes in the nanosecond range, while all previously reported exciplexes have a much longer-lived emission in the range of 50–100 ns (see ref. [6] and references therein). An air-equilibrated alkane solvent contains about 2 × 10^−3^ M dissolved oxygen [106] that quenches the long-lived emission, e.g., the experimentally measured fluorescence lifetime of naphthalene in air-equilibrated cyclohexane solution is ca. 16 ns as compared to the intrinsic fluorescence lifetime of 96 ns. The quenching is accompanied by suppression of the emission band, and ref. [6] reported that in comparison with degassed samples, the nominally identical air-equilibrated samples of naphthalene in *n*-dodecane are quenched by a factor of 5.4 for optical excitation and 6.7 for X-irradiation (integral quenching over the emission band of naphthalene in the range 320–400 nm, higher quenching under X-rays is due to the additional quenching of radical ionic processes). This calibration by the known compound helps avoid the unknown detailed mechanism and rate constants for quenching with oxygen in our specific system. On the other hand, the emission bands corresponding to the short fluorescence lifetimes of ca. 1 ns are virtually immune to the presence of oxygen and remain practically unchanged in an air-equilibrated sample.

Figure 4 shows the spectra of X-ray-induced luminescence for the mixtures of the four compounds with DMA in degassed and air-equilibrated *n*-dodecane solutions. Sample compositions are identical to those used in Figure 3 and, in fact, the air-equilibrated samples are the same solutions put in similar tubes from the same batch, but not subjected to the freeze–pump–thaw degassing and sealing, and just closed with Viton stoppers. The figure shows three spectra for each compound. The black curves for the degassed samples are the spectra from Figure 3. The red curves are spectra from the air-equilibrated samples of the same nominal composition, if needed slightly adjusted in magnitude to match the shorter wavelength bands. It can be seen that in all three cases, the exciplex emission band completely disappeared, while the shorter wavelength band remained nearly intact (seen from the similar signal-to-noise ratios) due to short lifetimes of this emission. The third, blue curves in each panel show the differences between these pairs of spectra, and subtraction nearly eliminates the shorter wavelength band and leaves the isolated exciplex emission band where it was initially present. This figure also faithfully represents the possible artifacts for such type of experiment for the rather weak spectra: the perfect match was achieved for **3**, in the other three cases, there is some residual mismatch at the shorter cut-off edge attributed to the slight variation in the wall thickness of hand-made ampoules from uncalibrated tubes made of material with a very sharp cut-off absorption. The spectra for the largest oligophenylene **4** show a longer tail of the local emission band at the spectrum for the degassed sample that is not present at the spectrum for the air-equilibrated sample, and this is attributed to working at the solubility threshold of the compound and the earlier-mentioned appearance of the traces of suspended solid compound after freezing and sealing the sample. The very low absolute concentration of **4** in solution and the resulting low probability of forming A^−•^ to participate in radical ion pair recombination is also a factor that exaggerates the imperfections of this procedure.

The described quenching with dissolved oxygen, in addition to providing the isolated exciplex emission bands, can also be used to obtain an estimate of the emission lifetime for the exciplexes. While in the earlier publication [6] the quenching was partial and required a more careful processing to obtain the estimated exciplex lifetimes of ca. 50–70 ns, for the compounds of this work, the exciplex band is completely eliminated by dissolved oxygen. Setting for an estimate that this corresponds to a 10-fold quenching and that a 100 ns species is quenched by a factor of ca. 5, we obtain the lower boundary for the exciplex emission lifetime of at least 200 ns. This is the longest time reported for radiation-generated exciplexes so far and may be the longest reported time for any exciplex. It may well be that exciplexes produced optically from energy donors with very long-lived excited states, such as pyrene, have a rather long-lived exciplex emission, but these experiments are necessarily performed in the conditions of eliminated quenching. On the other hand, there seems to be no straightforward correlation between the lifetimes of local excited states and exciplexes for the same system, e.g., exciplexes naphthalene/DMA and *p*-terphenyl/DMA have estimated emission lifetimes of 54 and 65 ns, respectively, with local emission lifetimes of 96 and 0.95 ns, respectively [6]. It is not yet clear what determines the exciplex emission lifetimes and how they can be explained, but the possibility of X-ray-induced generation of a wide range of recombination exciplexes from luminophores with arbitrary lifetimes of local emission and no restriction of the absorption spectra may help obtain more experimental data to form the basis for such an understanding. In any case, the long lifetimes of exciplex emissions is a factor that needs to be taken into account, as the substitution of a short-lived local emission with a longer-lived and spectrally shifted exciplex emission may drastically change emission properties of luminophore systems, especially if the emission spectrum and/or kinetics are not controlled, which may be critical, e.g., in the time-resolved studies or OLED applications, limiting time resolution or attainable energy throughput, respectively.

### 2.5. Magnetic Field Sensitivity of the Exciplex Emission Band

A sure sign of the recombination pathway of exciplex formation in the condition of radiation chemistry generation is the dependence of its emission intensity on the applied external magnetic field [3,5]. This is explained by the magnetic field sensitivity of singlet excited-state generation upon the recombination of radical ion pairs that, upon radiation generation, are partially spin-correlated. While a detailed description of this so-called “radiation spin chemistry” should be sought in specialized publications elsewhere [107,108,109], a brief explanation is as follows. The cascade of radiation chemistry processes in alkanes involves paramagnetic species, most importantly, radical ions, and thus depends on their electron spin state. The elementary processes such as initial ionization, charge capture and pair recombination all conserve the spin state. Since the radical ions are initially generated by the ionization of a closed-shell solvent molecule, the primary pair of the solvent radical cation and free electron appears from the same molecule (is geminate) simultaneously and inherits its singlet electron spin state, i.e., the radical ions are created as spin-correlated singlet radical ion pairs. Charge capture by acceptors involves the interaction of a radical ion with a closed-shell molecule, and the spin state is passed on to the newly formed radical ion simultaneously with charge transfer. The recombination of oppositely charged radical ions proceeds by a distant electron transfer and, in warm alkane solutions, occurs at the first sufficient approach to each other, irrespective of the collective electron spin state of the pair, forming either singlet state or triplet excited state of one of the pair partner parent molecules, depending on the pair multiplicity at the instance of recombination. In the traditional paradigm of the radiation spin chemistry, the long-lived triplet excited states deactivate radiationlessly due to their long lifetime, and singlet excited states produce the observed recombination or delayed fluorescence. The net result is that the primary radical ion pair is initially created in a singlet electron spin state and, after several tens of nanoseconds, the secondary radical pair recombines and produces a singlet-excited state only if it were singlet at the moment of recombination.

During the period from generation to recombination, the spin state of the pair can change due to magnetic interactions, most importantly due to hyperfine interactions with magnetic nuclei in the radical ions such as protons or fluorines, and interaction with an applied external magnetic field, if any. Therefore, an external magnetic field can modulate the probability of creating a singlet excitation upon recombination. In the most basic picture, hyperfine interactions are strong enough to spread the pair over all available electron spin states out of one singlet and three triplets. In the zero magnetic field, all four states are degenerate and the initially singlet pair is spread over all four states, and in a sufficiently high field, triplet states T_+_ and T_−_ are split away from singlet and T_0_ by Zeeman interaction, and the initially singlet pair spreads over just two states out of the four, which increases the probability of its recombination in singlet state relative to the no-field case. This is certainly an oversimplification, and in reality, the situation can be much more interesting [110,111,112], especially for pairs comprising fluorinated radical anions often having very large hyperfine coupling constants [113,114,115,116], but this suffices to explain why an applied field of 200 Gauss, as used in this work, may be expected to change the intensity of emission.

Figure 5 shows the spectra of X-ray-induced luminescence for the mixtures of the four compounds with DMA in degassed *n*-dodecane solutions with the application of magnetic field. The pair of spectra shown for each compound correspond to the spectra taken simultaneously by field cycling between 0 and 200 Gauss at each point of the wavelength scan. The spectra were not adjusted in any way, except for the subtraction of the identical baseline level and multiplication by the identical spectral sensitivity curve (see Section 3). Turning first to Figure 5b for perfluoro-*para*-terphenyl, it can be immediately seen that the exciplex emission in the region of 500 nm is enhanced by the application of the magnetic field of 200 Gauss, while the local emission band at ca. 340 nm is not visibly sensitive to the magnetic field. The sense of the effect corresponds to an increasing intensity with the field applied, and is consistent with the generation of exciplex emissions via the singlet-state recombination of the pair. The size of the effect is about 15%, which is very large: the maximum magnetic field enhancement possible in the conditions of this experiment is about 20%, and was only achieved in the exciplex emission band in specially developed systems based on substituted diphenylacetylenes [5]. The normal size of the magnetic field effect in similar experiments without spectral selection is of the order of 1%, which almost invariably necessitates the use of magnetic field modulation techniques with lock-in detection, similar to CW ESR.

The attainable size of the effect is limited by the fraction of spin-correlated radical ion pairs, which is always lower than 100% in reality, due to the clustered formation of ionizations in irradiated media, when several overlapping pairs are created within a limited spatial region and cross-recombinations between the partners of different overlapping pairs that are not correlated become possible. The very large magnetic field effect in the exciplex emission band is ensured by spectral selection, as this band is generated only by recombination of the secondary radical ion pair, and thus bears the maximum possible magnetic field sensitivity. The local emission band, on the other hand, is excited by a multitude of recombination and excitation harvesting processes that are possible in a sample irradiated by a high-energy ionizing radiation, and is therefore substantially diluted by background channels of fluorescence excitation. In other words, the useful emission related to the magnetic field sensitive recombination has mostly been shifted to a separate spectral band of exciplex emission, where it produces a close-to-theoretically-possible magnetic field effect. In paper [5], this phenomenon was actually reversed to experimentally determine the fraction of spin-correlated radical ion pairs in a range of liquid alkanes, from the size of the magnetic field effect in the exciplex emission band for a specially tailored probe pair, solving a long-standing problem in radiation chemistry.

Turning to the other panels of Figure 5, it is difficult to directly see any magnetic field sensitivity in the exciplex emission bands that are too weak to show separation of the noisy traces with and without an applied magnetic field. These spectra already required about 4 h of equipment time each, and it would take efforts beyond reason to significantly improve the experimental signal-to-noise ratio by further spectrum acquisition. However, a convenient tool to uncover any systematic differences between similar noisy spectra is integration over the bands of interest. Table 2 shows the results of such an integration for all the spectra of Figure 5, separately, in a local emission band and exciplex emission band for **2**–**4**, and over the entire spectrum for **1**, where no exciplex band is present. In all three cases, the exciplex emission band shows a magnetic field effect significantly exceeding the background effect in the local emission band, which confirms the recombination nature of exciplex emission for all three oligophenylenes, even when not immediately obvious directly from the experimental spectra.

### 2.6. Computational Methods

It would be very interesting to reproduce the observed exciplex formation for **2**–**4** in calculations, and understand why **1** does not seem to produce an exciplex such as the larger oligophenylenes, and has special electronic properties. Since the compounds **2**–**4** behave similarly in all our experiments, we chose to compare the smallest molecule of this series, **2**, with **1**. Quantum chemical calculations for **1** and **2** in neutral, radical anion and first excited states, and DMA in neutral and radical cation states were carried out using the Gaussian 16 [117], ChemCraft software to visualize the calculation results [118]. The wB97X-D3 [119,120] hybrid functional, including long-range and dispersion corrections, and standard 6-311G++(d,p) basis set were used. The geometry of the compounds in singlet and doublet states was optimized using restricted (R–) and unrestricted (U–) DFT approaches. The DFT mesh, the convergence criteria for the self-consistency procedure and the geometry optimization were set to defaults. Symmetry restrictions were not applied in the calculations. Hessians did not have negative frequencies.

First, neutral, radical anion, and the first excited states of **1** and **2** and neutral and radical cation states of DMA, were optimized. Then, starting with the optimized geometric structure in the corresponding electronic state, single-point TDDFT calculations were performed to obtain the energies of vertical electronic transitions. We used the fact that the TDDFT transition energies for the equilibrium ground state correspond to the absorption spectra, while TDDFT transition energies for the equilibrium excited state correspond to the fluorescence spectra. C-PCM [121,122] was included in all calculations to take into account the implicit solvent model, for which heptane was used. The TDDFT approach has been previously successfully applied to model the structure of exciplexes, in which the acceptor (in our case **1**–**4**) was in an excited state, while the donor (DMA) was in the ground state [123,124,125,126,127,128], while the chosen functionals, including the long-range and dispersion corrections, demonstrated a good agreement between the experimental and calculated spectra [124,126,127]. According to the robust modeling strategy described in ref. [127], to determine the equilibrium geometry and the corresponding transition energies, the exciplex should be optimized in the excited state. This is a major and rather expensive computational problem, and, as can be seen from Figure 4 and Figure 5 presented in ref. [127], the optimization leads to the acceptor bending deformation. However, we believe that the distance between the donor and acceptor plays a more important role in the appearance of long wavelength fluorescence lines due to the formation of an exciplex. Therefore, for this work, we chose a less computationally expensive and more qualitative approach, and varied the donor–acceptor distance from 2.0 to 10 A, performing single-point calculations of the transition energies to estimate the distance at which an exciplex emission appears, interpreting it as the sign of the exciplex formation and comparing the predicted emission wavelength with the experimentally observed position of the exciplex emission band.

### 2.7. Geometric and Electronic Structures of ***1***, ***2*** and DMA

To check the possibility of exciplex formation in these systems and understand the origin of the special behavior of perfuorobiphenyl in comparison with its larger siblings, DFT calculations for **1**, **2** and DMA were undertaken, as described above. First, we performed a test calculation of the perfluorobenzene radical anion with the chosen methods. A good agreement with results of ref. [128] was obtained, where the B3LYP/6-311G+(d) approach confirmed the nonplanar structures, including a chair structure with a puckered carbon-ring, a structure arising from the electrostatic repulsion of the fluorine atoms. In [129], several perfluorinated aromatic molecules (benzene, naphthalene and anthracene) were investigated in both neutral and radical anion forms at the B3LYP/DZP++ level. The authors dismissed the planar structures of perfluorinated benzene (D_6h_) and naphthalene (D_2h_) radical anions as having imaginary vibrational frequencies, while nonplanar structures, (C_2v_) and (C_2h_), respectively, did not have imaginary frequencies.

A similar result was obtained earlier in the calculations of perfluorobiphenyl radical anion with geometry optimization at the PBE/6-31G+(d) level in [130], where a nonplanar structure of fluorine atoms with respect to the carbon ring was also observed and confirmed by Optically Detected ESR experiments. The structural motility of polyfluorinated radical anions is in fact their general trait, often seen in specialized experiments and recovered in the highest-level calculations that aim at reproducing experimentally obtained hyperfine coupling constants [131,132,133,134,135]. In our calculations, the optimized structures of pefluorobiphenyl and its radical anion are planar for all fluorine atoms relative to their respective carbon ring (Appendix A), and for the deformed planar structure of fluorine atoms, the process of optimization invariably returned the fluorine geometry to planar. The dihedral angle between the aromatic rings of the perfluorobiphenyl anion radical 43.2° is in good agreement with the value of 45°, obtained in [130]. However, the optimization of the perfluorobiphenyl structure in the first excited state produced out-of-plane deviations of one C–F bond in one of the rings (dihedral angle ∠2-3-4-F4 = 139.9°).

Upon passing to perfluoroterphenyl **2**, distortions for fluorine atoms are no longer observed for radical anion nor the excited state (Appendix A). Comparing the dihedral rotation angles of aromatic rings for **1*** and **1**^•−^, it should be noted that the dihedral angle for **1*** (59°) differs markedly from the dihedral angle for **1**^•−^ (43°). In the case of **2*** (43°) and **2**^•−^ (47°), the difference is not so large and is in the opposite direction. Distortion of the planar structure of fluorines in **1*** is probably an additional factor leading to a larger dihedral angle. Other geometric characteristics in the respective pairs (**1**–**2**, **1**^•−^–**2**^•−^, **1***–**2***), such as bond lengths and bond angles in aromatic rings and structures of fluorine atoms, do not differ significantly between **1** and **2**. This, however, is not the case for the magnitudes of charges on the corresponding carbon and fluorine atoms. Larger charge values for all forms of **1** indicate a greater electrostatic interaction than in all their counterparts of **2**. Appendix A also provides geometric and electronic structure for DMA and its radical cation. The calculated and experimental absorption/emission spectra for DMA can also be found in Appendix A.

Figure 6 shows the calculated absorption and luminescence spectra for **1** and **2**, and their comparison with experimental spectra from Figure 1 and Figure 2 above. Qualitatively, all features of the experimental spectra are reproduced, but there is a systematic underestimation for the positions of all spectra maxima (~25 nm) produced by the used wB97X-D3/6-311G++(d,p) method. Similar underestimates were noted earlier by the authors of ref. [127]. We believe that such a systematic shift in the range of 20–30 nm is acceptable and gives us confidence in the applicability of the chosen calculation method. The calculations faithfully reproduce the doublet nature of absorption in **1,** with a reasonable ratio of oscillator strengths and a much longer wavelength emission of **1** in comparison to **2**, i.e., the spectral features that had set perfluorobiphenyl apart from the larger oligophenylenes **2**–**4** in their optical characterization above. Overall, it can be inferred that the removal of “inner” phenylene upon going from perfluoroterphenyl to perfluorobiphenyl produces more stressed structures with out-of-plane distortions in the excited state, which become even more stressed for hexafluorobenzene. Reversing this argument, insertion of (one or more) inner phenylenes between the terminal phenyls produces more relaxed structures of all the species. This presence/absence of the inner phenylene qualitatively explains the difference between perfluorobiphenyl **1** and larger perfluorinated oligophenylenes **2**–**4** in their optical properties related to the excited state, that are directly related to exciplex formation.

### 2.8. Formation of an Exciplex between ***1*** or ***2*** and DMA

As mentioned in the description of the calculation methods, a consistent approach to exciplex modeling would require the optimization of the structure of intermolecular complex in the excited state, but this is beyond the scope of this work, and we shall only build a reasonable model of the complex from the optimized structures of the excited acceptor and the ground-state donor, and vary the donor–acceptor distance. As authors of [127] note, the appearance of the exciplex emission line, its position and intensity depend on the relative positioning of the donor and acceptor. It is natural to assume that to maximize the interaction of the nitrogen lone pair of the donor with the π-system of the acceptor, DMA should be positioned with nitrogen opposite the center of an aromatic ring of **1*** or **2***. Interaction of hydrogens of DMA methyls with fluorines, due to the nonplanar structure of perfluorophenyl rings, should also be taken into account, as they may form hydrogen bonds. Based on these considerations, we built the initial structures of complexes DMA-**1*** and two variants of DMA-**2***, shown in Figure 7. The planes of the relevant aromatic rings of **1**/**2** and DMA were held parallel to each other. The two structures of **2** with DMA over outer/inner rings, in comparison with DMA over the ring of **1,** were taken to check how the basic idea about relieving the stress in an excited state by inserting an inner *para*-phenylene moiety can be extrapolated from single molecules to exciplexes.

To check the influence of the distance from the donor to the acceptor, we ran a series of single-point calculations of the fluorescence spectra of such complexes with varied distances. Juxtaposition of calculated and experimental emission spectra for a single molecule/exciplex of **1**, **2** and DMA, and the assignment of transitions to local or exciplex emission, are given in Appendix A. Figure 8 shows the distance dependence of the total energy of the system, E_0_, wavelength of luminescence of the excited acceptor (**1**/**2**), λ_l_, wavelength of exciplex luminescence, λ_ex_, and their intensities (oscillator strengths, f_osc_). All three complexes have a potential well with a characteristic energy minimum in the vicinity of about 3.2A, implying the formation of a bound state that would only be more stabilized by full optimization. For all complexes, the position and intensity of the exciplex emission line strongly depend on the distance from the donor to the acceptor. Moreover, the line of exciplex disappears at distances above 5A for all complexes. On the other hand, the positions and intensities of the lines of local luminescence of acceptors **1*** and **2*** depend on the distance to a lesser degree, and for distances exceeding 2.8A, the luminescence wavelengths and intensities do not change and are close to values indicated in Figure 6 for the emission of isolated molecules.

The emission line of exciplex DMA-**1*** in the potential well minimum has a wavelength of 329 nm and an intensity of 0.03. It happens to be very close and at the blue side to the (calculated) emission line of **1*** (λ_l_ = 345 nm, f_l_ = 0.052), which may be the reason why no separate line of the exciplex for **1** has been observed in the experimental spectra of X-ray-induced luminescence, while the predicted bound state is similar to **2**. Exciplex emission lines for the two considered cases of DMA-**2*** (DMA opposite to the inner or outer rings) at the potential well minimum are significantly red-shifted with respect to the local emission of the acceptor, and have the following wavelengths and intensities: 393 nm (0.005) and 365 nm (0.081), respectively. Additionally, it should be noted that a small movement toward lower distances rapidly red-shifts the exciplex emission line closer to the experimentally observed range. In the case of DMA opposite to the inner ring of **2***, the predicted line position is closer to the experimentally observed position of the exciplex emission band, but it has a lower intensity than in the case of DMA at the outer ring, which in turn predicts a smaller red shift of the exciplex emission line relative to the local emission. In line with the experimental results and with the single-molecule calculation described above, it can be observed that the DMA-**1*** exciplex shows spectral behavior completely different from that of both exciplexes of DMA-**2***, supporting the key importance of relieving the structural stress in the excited state, rather than the distinction of whether DMA is positioned over the outer/inner ring of a three-ring structure. The unexpected positioning of the exciplex emission line to the blue of the local emission for the DMA-**1*** exciplex may be explained by the same underlying idea of relieving the stress when it becomes possible, as follows. The unusually red-shifted local emission of **1** is attributed to its stressed-excited state with an overcrowded electron density. When DMA is moved close enough to participate in the sharing electron density, the stress is relieved as it was relieved after the insertion of the inner phenylene. This interaction also gives rise to exciplex formation, with its emission band located where it may be expected for a would-be relieved structure having a not so red-shifted local emission. Overall, we believe that our simple proof-of-principle calculations turned out surprisingly successful, qualitatively reproducing the main features of local emission, predicting exciplex formation and emission wavelength, and, most importantly, pinpointing the differences in the local emission and exciplex formation behavior of perfluorobiphenyl in comparison to perfluoroterphenyl (and larger oligophenylenes) that were indeed observed experimentally.

### 2.9. Conclusions and Outlook

We report an unprecedented exciplex formation between a series of perfluorinated *para*-oligophenylenes C_6_F_5_-(C_6_F_4_)_n_-C_6_F_5_ (*n* = 1–3) with *N*,*N*-dimethylaniline (DMA) in degassed X-irradiated *n*-dodecane solutions. This experiment takes advantage of the ability to produce exciplexes from luminophores with an arbitrarily short local excited-state lifetime and no restrictions on the UV-Vis spectra of the participating molecules under X-rays, where an efficient assembly of exciplexes proceeds via the recombination of radical ion pairs, which delivers the two partners close to each other and ensures a sufficient energy deposition. The systems of this work, indeed, have too-short fluorescence lifetimes (ca. 1.2 ns) and completely overlapping UV-Vis absorption spectra of the two components, with molar absorption coefficients of 2.7–4.6 × 10^4^ M^−1^cm^−1^, which precludes a standard photochemical exciplex formation pathway via selective optical generation of a local excited state of the donor and its bulk quenching by the acceptor. The paper discusses the underlying principles and available experimental choices for the generation and experimental study of such recombination exciplexes in sufficient detail, including the additional options for the experimental isolation of the exciplex emission band by quenching with dissolved oxygen, and the magnetic field sensitivity of the exciplex emission band inherited from the magnetic field sensitivity of the recombination of spin-correlated radical ion pairs. Exciplex formation in such systems is further explored by DFT calculations. These first exciplexes from fully fluorinated compounds show the largest known red shift of the exciplex emission from the local emission band, suggesting the potential of perfluorocompounds for optimizing optical emitters. Their most promising applications may be in the field of OLEDs [136,137,138,139,140,141,142,143,144,145,146], given that the radiation chemical pathway of exciplex generation closely mimics electroluminescence generation in OLEDs. The described solution–phase experiment may also be a useful alternative to the full-cycle fabrication of trial devices for routine testing, while the encountered difficulties with limited solubility in nonpolar liquids are absent for solid-state devices often using polymeric components with close-to-zero solubility. From a farther perspective, the availability of a simple experiment for the generation of isolated molecular exciplexes in solution from virtually any energetically and sterically suitable donor–acceptor pair, together with the explosive development of computational quantum chemistry and immediate practical potential for the application of the exciplexes in organic optoelectronics of today, may rekindle new interest in objects first considered 60 years ago.

## 3. Materials and Methods

The target perfluorinated *para*-oligophenylenes C_6_F_5_-(C_6_F_4_)_n_-C_6_F_5_ (*n* = 1–3) were prepared following well-known procedures. An ampoule was charged with 4.96 g (16.11 mmol) 1,4-dibromotetrafluorobenzene, 7.91 g (32.03 mmol) bromopentafluorobenzene and 10.14 g (159.57 mmol) copper. The ampoule was sealed, placed in a protective metal jacket and heated at 240–250 °C for 12 days. After the completion of the reaction, the ampule was cooled to RT and opened, and the mixture was extracted with CHCl_3_ (2 × 30 mL). The combined extracts were filtered through a layer of silica gel and the filtrate was evaporated. The residue was sublimed under reduced pressure (~12 mm Hg) step wise to separate the products. Sublimation at 120 °C produced 2.15 g of decafluorobiphenyl **1** (*n* = 0, 40% yield). Sublimation at 140 °C produced 0.42 g perfluoro-*para*-terphenyl **2** (*n* = 1, 5% yield). Sublimation at 170 °C produced 0.45 g perfluoro-*para*-quaterphenyl **3** (*n* = 2, 4% yield). Finally, sublimation at 210 °C yielded 0.11 g perfluoro-*para*-quinquephenyl **4** (*n* = 3, 1% yield). The yields were based on bromopentafluorobenzene. The identity and purity of the target compounds **2**–**4** were confirmed by ^19^F NMR spectra (Bruker AV-300 (282.40 MHz) in CHCl_3_–CDCl_3_), which were consistent with the literature data [147,148,149]. Compound **1** was also available from alternative sources (Decafluorobiphenyl 99%, Avocado Research Chemicals Ltd., London UK), and this was used in the study. Further, 1,4-Dibromotetrafluorobenzene (97%) and bromopentafluorobenzene (98%) were purchased from P&M Invest, Moscow, Russia. Copper was preliminarily calcined at 310–320 °C for 16 h in a sealed ampule. DMA (99%, Sigma-Aldrich, Burlington, MA, USA) was distilled over zinc powder and stored in the dark, in a freezer.

*n*-Dodecane (reagent grade, TU6-09-3730-74) was purchased from Reakhim, Moscow, Russia, and additionally purified as follows. To 1 L of dodecane in a 2 L flask, 100 mL of H_2_SO_4_ and 3 g of Ag_2_SO_4_ were added, and the solvent was stirred for 4 h at RT. The aqueous layer turned brown due to the products of oxidation of unsaturated impurities that were extracted into the aqueous phase. The aqueous phase was removed and the treatment was repeated for 4 more cycles, until the coloration disappeared. Then, the solvent was washed 4 times with distilled water, until a neutral pH of the wash, dried over CaCl_2_ and passed through a 18 cm column with activated Al_2_O_3_. The attained purity was checked by UV-Vis and fluorescence measurements (see Appendix A) that demonstrated traces of impurities with OD ca. 0.15 in the range 225–275 nm (1 cm optical path length), corresponding to the concentration of impurities at the level of 5 × 10^−6^ M (assuming typical molar absorption coefficients of 2–3 × 10^4^ M^−1^cm^−1^), with intrinsic fluorescence in the range of 290 nm, with a fluorescence lifetime of ca. 10 ns. While inconvenient for optical fluorescence measurements, this is acceptable for radiation chemistry studies with 10^2^ to 10^4^ higher concentrations of additives. All optical studies were performed using spectrophotometry-grade acetonitrile (Cryokhrom, St-Petersburg, Russia). The fluorescence quantum yield standard was prepared in UV–IR–HPLC-grade cyclohexane (PanReac AppliChem, Darmstadt, Germany).

All optical measurements were performed in 1 cm cuvettes. UV-Vis spectra were taken on a Shimadzu UV-2401PC UV-Vis Recording Spectrometer (Shimadzu, Kyoto, Japan), with an optical slit width of 1 nm. Fluorescence-related measurements (emission and excitation spectra, and emission kinetics) were performed with an Edinburgh Instruments FLS900 Fluorescence spectrometer (optical slit width of 2/2 nm for emission and excitation spectra). UV-Vis spectra for samples used in fluorescence characterization (Appendix A) were taken on a HP/Agilent 8453 UV/Vis Spectrophotometer. Quantum yields were determined using naphthalene in cyclohexane (φ = 0.23 [88]) as a standard. The relatively long intrinsic fluorescence lifetime of naphthalene, 96 ns, is a complicating factor, as emission is significantly quenched by oxygen dissolved in solvent. However, we took advantage of the previously experimentally established fact [6] that equilibrating with air quenches naphthalene emission in *n*-dodecane by a factor of 5.4 (integral over emission band), as compared to the same sample degassed by the freeze–pump–thaw cycles on a forepump vacuum post, and used 0.23/5.4 = 0.043 as the standard quantum yield from a solution of naphthalene in cyclohexane (not degassed and equilibrated with air). The measured fluorescence time of naphthalene in air-equilibrated cyclohexane of ca. 16 ns was also consistent with this estimate. From a practical viewpoint, this provided a convenient standard for the shorter wavelength range (excitation wavelengths 250 nm and 270 nm were used), as the common reference luminophores usually have a longer wavelength absorption beyond 300 nm.

X-ray-induced luminescence experiments were performed on a home-built magnetically affected reaction yield (MARY) spectrometer with the spectral resolution of luminescence described in an earlier publication [1]. The samples, having volume of 250 μL, were placed in round, thin-walled ampoules, with an outer diameter of 5 mm made from molybdenum glass, which was earlier found to produce virtually no intrinsic luminescence under X-irradiation. Unless stated otherwise, the samples were degassed using three freeze–pump–thaw cycles on a forepump vacuum post and sealed off with a torch. To further suppress the residual background X-ray-induced emission from the sample wall, the samples were placed in a blackened, thick-walled lead jacket in the form of a Pb tube, with an inner diameter of 5 mm and an outer diameter of 9 mm, with two parallel vertical slits 10 mm high and 2 mm wide cut in the wall at locations 90 degrees apart from each other. This produced a 3D-masked orthogonal geometry for sample irradiation and light collection, completely decoupling the observation optical port from the parasitic background emission of the exposed tube wall at the X-ray excitation port. The sample was exposed to the incident X-ray beam (unfiltered bremsstrahlung from a CW X-ray tube 2,5BSV-27-Mo, Svetlana, St Petersburg, Russia, 40 kV × 20 mA, sample to anode distance of 210 mm, beam height of 10 mm) and to the light-collecting optics of the detection system comprising a quartz optical imaging system, a grating monochromator (MDR-206, LOMO Photonics, St Petersburg, Russia, objective focus length 180 mm, grating 1200 lines per mm, inverse linear dispersion 4.3 nm mm^−1^) with slits set to 2.2 mm/2.2 mm (spectral resolution about 10 nm) and a Hamamatsu H10493-012 photosensor module. All experiments were performed within ambient conditions, without temperature control.

Each reported X-ray-generated luminescence spectrum is an average over 6 wavelength scans (512 wavelength points in the range of 250 to 1000 nm) of ca. 18 min each, with a total spectrum acquisition time of ca. 2 h. The longer wavelength end was used to determine the baseline level that was subtracted, and the spectra were then corrected for spectral sensitivity by multiplying by the calibration curve, determined earlier using a calibrated “black body” spectral lamp with a known temperature. The spectra were then cut at 650 nm, after which second-order diffraction phantoms were observed from intense local emissions in the region of 350 nm. For magnetic field experiments, the wavelength scan was augmented with magnetic field cycling between 0 and 200 Gauss at each wavelength point during the scanning, simultaneously producing a pair of spectra with and without applied fields. These spectra were taken in the limited wavelength range (300 to 600 nm) and at 256 points per scan due to slower operation in this mode (ca 40 min per such scan), averaging over 6 scans, and the total spectrum acquisition time was ca. 4 h. The spectra were also corrected for baseline and spectral sensitivity. All spectra were recorded in nominally identical conditions, intensities were given as arbitrary units corresponding to the output signal of the detector after multiplication by the universal spectral sensitivity curve, and are consistent for all spectra. The y-axes can be directly compared between different spectra.

## Figures and Tables

**Figure 1 ijms-24-07568-f001:**
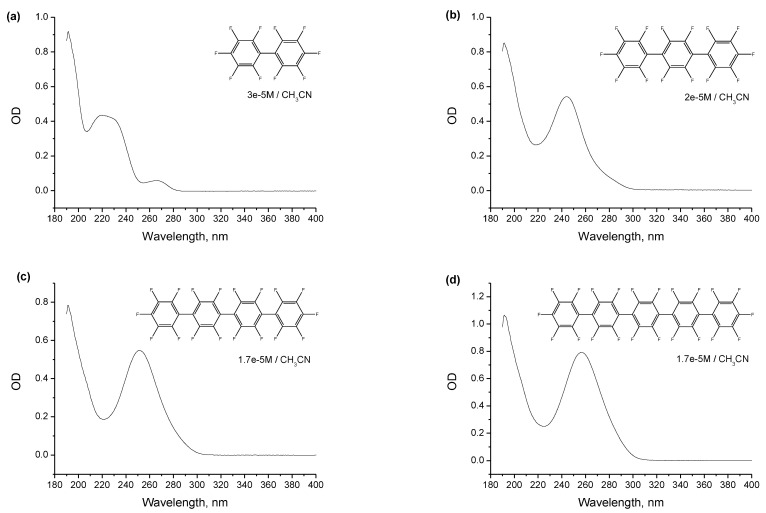
Absorption spectra of the four compounds ((**a**)—**1**, (**b**)—**2**, (**c**)—**3**, (**d**)—**4**) in acetonitrile at indicated concentrations. Positions of the maxima and molar absorption coefficients at the maxima are collected in Table 1.

**Figure 2 ijms-24-07568-f002:**
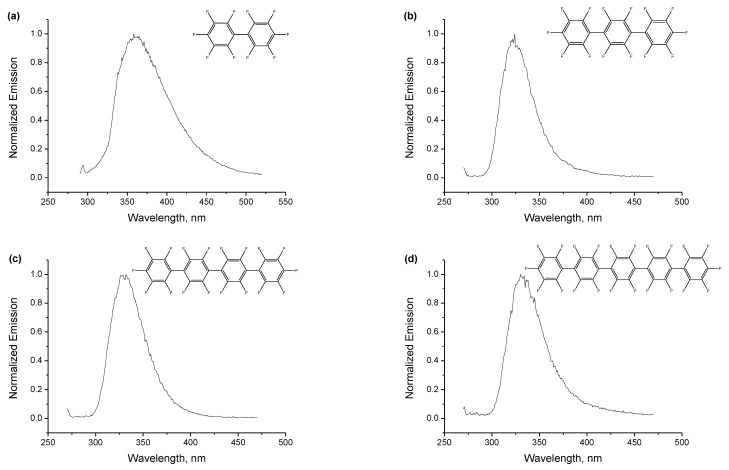
Normalized emission spectra of the four compounds ((**a**)—**1**, (**b**)—**2**, (**c**)—**3**, (**d**)—**4**) in acetonitrile. Positions of the emission maxima are collected in Table 1. Excitation at 270 nm for **1** and 250 nm for **2**, **3**, **4**. The raw emission spectra are given in the Appendix A.

**Figure 3 ijms-24-07568-f003:**
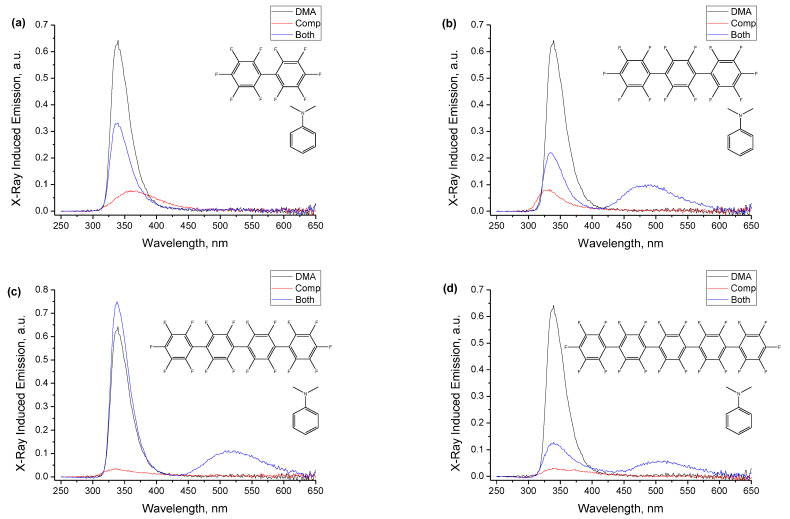
Spectra of X-ray-induced luminescence for the four compounds ((**a**)—**1**, (**b**)—**2**, (**c**)—**3**, (**d**)—**4**) in degassed *n*-dodecane solutions. Each panel shows a group of spectra from three samples: only DMA (black curve), only the compound (red curve), and their mixture at the same concentrations of the components (blue curve). The DMA concentration in all samples is 10^−2^ M. Concentrations of compounds (with the exception of **1**) are limited by their solubility in *n*-dodecane and are: 5 × 10^−3^ M for **1**, 1.8 × 10^−3^ M for **2**, 4.4 × 10^−4^ M for **3**, 1.1 × 10^−4^ M for **4**.

**Figure 4 ijms-24-07568-f004:**
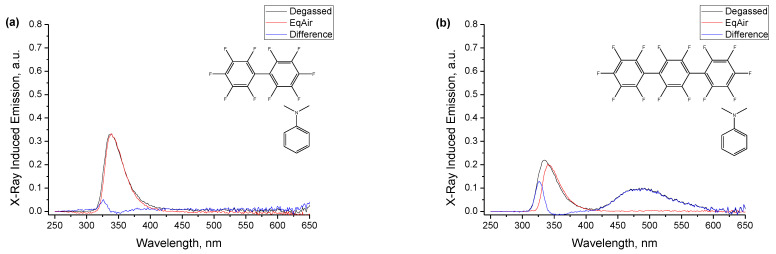
Spectra of X-ray-induced luminescence for the mixtures of the four compounds ((**a**)—**1**, (**b**)—**2**, (**c**)—**3**, (**d**)—**4**) with DMA in degassed and air-equilibrated *n*-dodecane solutions. Sample compositions are identical to those used in Figure 3. Each panel shows three spectra: from the degassed solution (black curve), from the same solution equilibrated with an ambient atmosphere (red curve, adjusted in magnitude to match the intensity of the short wavelength band of the local emission), and their difference (blue curve). A DMA concentration of 10^−2^ M in all samples, concentrations of compounds: 5 × 10^−3^ M for **1**, 1.8 × 10^−3^ M for **2**, 4.4 × 10^−4^ M for **3**, 1.1 × 10^−4^ M for **4**.

**Figure 5 ijms-24-07568-f005:**
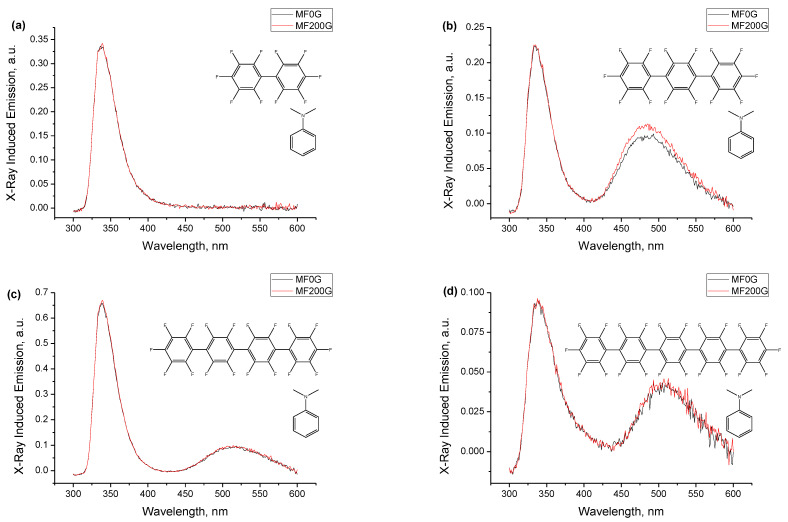
Spectra of X-ray-induced luminescence for the mixtures of the four compounds ((**a**)—**1**, (**b**)—**2**, (**c**)—**3**, (**d**)—**4**) with DMA in *n*-dodecane solutions with an applied magnetic field. Sample compositions are identical to those used in Figure 3. Each panel shows two spectra from the degassed solution taken by field cycling at each point of the wavelength scan: without the magnetic field (black curve) and in the applied field of 200 G (red curve). Magnetic field effects obtained by the integration over the suitable spectral ranges are collected in Table 2. A DMA concentration of 10^−2^ M in all samples, concentrations of compounds: 5 × 10^−3^ M for **1**, 1.8 × 10^−3^ M for **2**, 4.4 × 10^−4^ M for **3**, 1.1 × 10^−4^ M for **4**.

**Figure 6 ijms-24-07568-f006:**
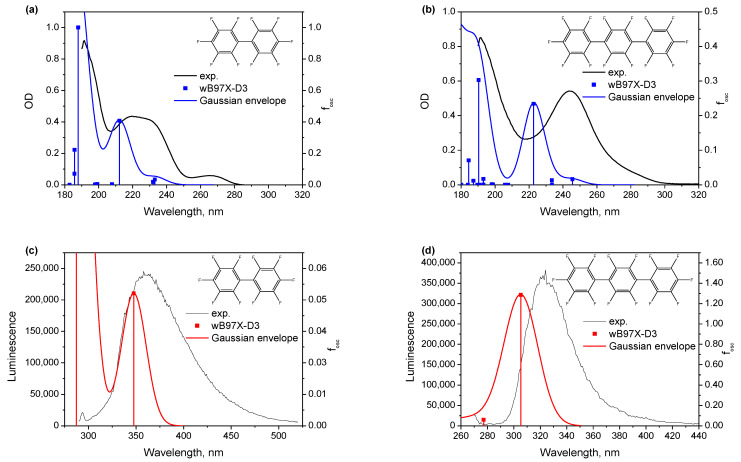
Calculated absorption lines for equilibrium structures of **1** (**a**) and **2** (**b**) in heptane (square points) and the corresponding experimental absorption spectra from Figure 1, and calculated absorption lines for equilibrium structures of **1*** (**c**) and **2*** (**d**) in heptane (square points) and the corresponding experimental luminescence spectra from Figure 2.

**Figure 7 ijms-24-07568-f007:**
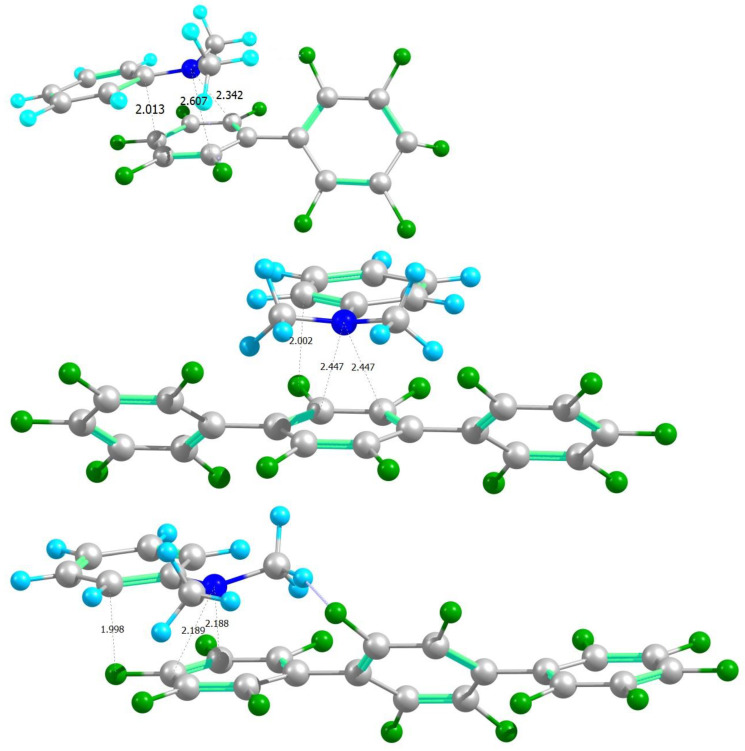
Top to bottom: structures of exciplexes formed by DMA over the outer ring of **1***, inner ring of **2*** and outer ring of **2***. The colors of the atoms are as follows: H—cyan, C—gray, N—blue, F—green.

**Figure 8 ijms-24-07568-f008:**
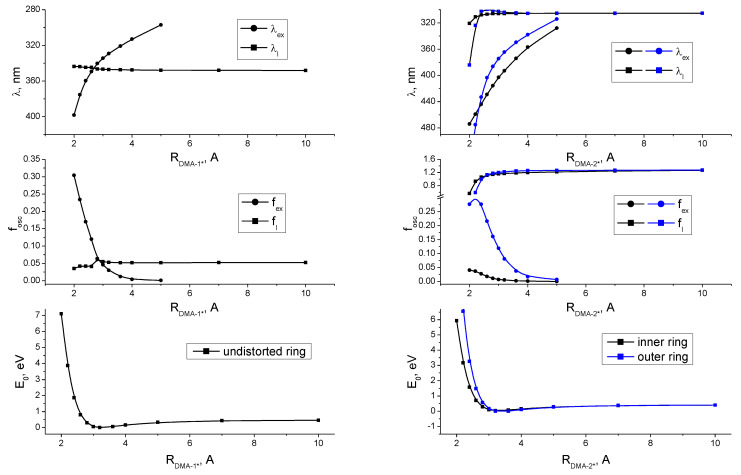
Total energy, E_0_; exciplex luminescence wavelength, λ_ex_, and its intensity, f_ex_; local luminescence wavelength, λ_l_, and its intensity, f_l_; as functions of the distance between DMA and **1*** (left panel) and DMA and **2*** (right panel). The minimum total energy is E_0 min_ = −49572.46 eV for **1**, and E_0 min_ = −66660.97 eV for **2**.

**Table 1 ijms-24-07568-t001:** Optical properties of perfluorinated *para*-oligophenylenes.

Compound	λ_max_, nm ^a^	ε, M^−1^cm^−1 b^	λ_em_, nm ^c^	φ ^d^	τ, ns ^e^
**1**	222/267	1.4 × 10^4^/0.2 × 10^4^	360	0.024	1.71
**2**	245	2.7 × 10^4^	324	0.043	1.23
**3**	251	3.2 × 10^4^	330	0.039	1.20
**4**	257	4.6 × 10^4^	332	0.035	1.17

**a** Positions of the maxima in UV-Vis absorption spectra; **b** molar absorption coefficients at the absorption maxima; **c** positions of the maxima in emission spectra; **d** fluorescence quantum yield; **e** fluorescence lifetime. All spectra taken in acetonitrile, for fluorescence measurements (**c**–**e**) excitation at 270 nm for **1** and 250 nm for **2**, **3**, **4**, based on the availability of EPLEDs used as a pulsed light source for time-resolved measurements.

**Table 2 ijms-24-07568-t002:** Magnetic field effect on local emission/exciplex emission bands.

Compound	Integration Range, nm ^a^	Area (0 Gauss) ^b^	Area (200 Gauss) ^c^	Area(200)/Area(0) ^d^	MFE, % ^e^
**1**	300–600	13.61	13.79	1.013	1.3
**2**	300–410 410–600	8.209 8.824	8.339 10.205	1.016 1.157	1.6 15.7
**3**	300–440 440–600	23.29 7.878	23.62 8.226	1.014 1.044	1.4 4.4
**4**	300–430 430–600	4.289 3.473	4.381 3.731	1.021 1.074	2.1 7.4

**a** Range of wavelengths in X-ray-induced luminescence spectra, over which the signal was integrated; **b** area under the curve (AUC) for the spectrum taken in the zero applied magnetic field; **c** AUC of the spectrum taken in an applied magnetic field 200 Gauss; **d** ratio of AUCs in the magnetic field/without magnetic field; **e** magnetic field effect defined as per cent increase in signal intensity in applied magnetic field.

## Data Availability

All data for this study are explicitly included in the main text and Appendix A available at MDPI.

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
