# Peer review of "Recombination of X-ray-Generated Radical Ion Pairs in Alkane Solution Assembles Optically Inaccessible Exciplexes from a Series of Perfluorinated para-Oligophenylenes with N,N-Dimethylaniline"

_ijms, 2023, doi:10.3390/ijms24087568_

Round 1
Reviewer 1 Report
This paper concerns the formation of exciplexes between perfluorinated p-oligophenylenes and N,N-dimethylaniline after recombination of radical ion pairs generated by X-ray irradiation. The authors have introduced a novel experimental method for investigating the character of electronic excited states in solution using X-ray ionization of solvent molecules. They have designed their experimental setup including a sample container, carefully selected target molecules and solvents, and thoroughly purified them for avoiding emission from impurities. They have found a long-wavelength emission band for a mixture of perfluorinated p-oligophenylenes and N,N-dimethylaniline in dodecane, separated clearly from local emission bands, and assigned it to an exciplex emission. The assignment was supported by magnetic field sensitivity experiments and TDDFT calculations. The authors’ experimental method is similar to the pulse radiolysis, which has been extensively used for investigating the nature of electronic excited states. In the authors’ method, however, the secondary radical ion species are generated only in the vicinity of the initially generated radical ions while high-energy electrons in the pulse radiolysis could cause complex ionization processes.
The experimental results strongly suggest the formation of exciplexes between DMA and perfluorinated oligophenylenes. The manuscript is concisely written so broad readership in the area of photochemistry as well as radiation chemistry will understand the significance of the authors’ study. The reviewer thinks, however, that the TDDFT calculation part can be improved for convincing readers of the exciplex formation. The detailed comments are listed below.
1. The authors assumed two structures for the DMA-2* exciplex: DMA over the inner and outer ring of 2*. If the DMA-2* exciplex with the latter structure is formed, it will show a similar spectroscopic character to that of the DMA-1* exciplex because DMA is interacted with the outer ring in both exciplexes. In the experimental results, however, 1 and the DMA-1* exciplex show spectral behavior totally different from that of the other exciplexes. The reviewer wonders what the authors intend by considering the DMA-2* exciplex with DMA over the outer ring.
2. It is unclear what transition the exciplex emission can be assigned to for Figure 8. Is it possible for the exciplex emission wavelength to be shorter than that of the local emission as shown in the top panel at the left-hand side of Figure 8? It will be helpful if the authors could prepare an energy diagram for explaining the transitions they have observed.
Author Response
This paper concerns the formation of exciplexes between perfluorinated p-oligophenylenes and N,N-dimethylaniline after recombination of radical ion pairs generated by X-ray irradiation. The authors have introduced a novel experimental method for investigating the character of electronic excited states in solution using X-ray ionization of solvent molecules. They have designed their experimental setup including a sample container, carefully selected target molecules and solvents, and thoroughly purified them for avoiding emission from impurities. They have found a long-wavelength emission band for a mixture of perfluorinated p-oligophenylenes and N,N-dimethylaniline in dodecane, separated clearly from local emission bands, and assigned it to an exciplex emission. The assignment was supported by magnetic field sensitivity experiments and TDDFT calculations. The authors’ experimental method is similar to the pulse radiolysis, which has been extensively used for investigating the nature of electronic excited states. In the authors’ method, however, the secondary radical ion species are generated only in the vicinity of the initially generated radical ions while high-energy electrons in the pulse radiolysis could cause complex ionization processes.
The experimental results strongly suggest the formation of exciplexes between DMA and perfluorinated oligophenylenes. The manuscript is concisely written so broad readership in the area of photochemistry as well as radiation chemistry will understand the significance of the authors’ study. The reviewer thinks, however, that the TDDFT calculation part can be improved for convincing readers of the exciplex formation. The detailed comments are listed below.
- The authors assumed two structures for the DMA-2* exciplex: DMA over the inner and outer ring of 2*. If the DMA-2* exciplex with the latter structure is formed, it will show a similar spectroscopic character to that of the DMA-1* exciplex because DMA is interacted with the outer ring in both exciplexes. In the experimental results, however, 1 and the DMA-1* exciplex show spectral behavior totally different from that of the other exciplexes. The reviewer wonders what the authors intend by considering the DMA-2* exciplex with DMA over the outer ring.
Yes, thank you for this question. This seems indeed the most plausible suggestion, if you directly consider exciplex formation. However, it happened that we had to start at an earlier stage, in part because we did not know how to approach the calculation of exciplexes and started with simpler things we could hope to manage. We got the clue for our idea to check the two structures of exciplex with 2 that turned out to be similar and both different from the exciplex with 1, from the spectral characterization of isolated molecules 1 vs 2-4 (see Figs. 1-2 of the manuscript), in which 1 behaved strange and different from the higher homologues. So we first tried to understand this, and calculated the structures and emission/absorption spectra just for isolated molecules. And found the non-planar fluorine arrangement in the excited perfluorobiphenyl, and reproduced the even more heavily distorted, puckered carbon ring structure, of the smaller hexafluorobenzene radical anion used as the well-known reference. Most satisfyingly, our (fairly simple) calculations reasonably reproduced the strange absorption/emission spectra, which let us correlate the “stressed” spatial structure for excited perfluorobiphenyl with the appearance of its longer-wavelength bands. And both the stressed structure and the strange spectra disappeared upon adding one internal phenylene upon passing to 2. Extrapolating this to exciplexes then, the principal difference seemed to be not whether DMA is positioned over an outer/inner ring, but rather whether how stressed is this ring. In 1 it is stressed, while in 2 the stress is relieved throughout the molecule and there is no stress either at outer or at inner ring, which produces similar exciplexes, both different from 1. This was the basic idea behind this consideration, and it worked surprisingly well. We described all this in the final paragraphs of Section 2.7 and Section 2.8 for isolated molecules and exciplexes, respectively. However, now rereading the text and contemplating your question, we decided to indeed add the following phrase before Fig. 7 to emphasize why such structures were considered: “The two structures for 2 with DMA over outer/inner rings in comparison with DMA over the ring of 1 were taken to check how the basic idea about relieving the stress in excited state by inserting an inner para-phenylene moiety can be extrapolated from single molecules to exciplexes.” Also the following sentence was added later to the last paragraph of Section 2.8 to reflect your observation: “It can be seen that, in line with the experimental results and with the single-molecule calculation described above, the DMA-1* exciplex shows spectral behavior totally different from that of both exciplexes of DMA-2*, supporting the key importance of relieving the structural stress in excited state, rather than the distinction of whether DMA is positioned over the outer/inner ring of a three-ring structure.”
- It is unclear what transition the exciplex emission can be assigned to for Figure 8. Is it possible for the exciplex emission wavelength to be shorter than that of the local emission as shown in the top panel at the left-hand side of Figure 8? It will be helpful if the authors could prepare an energy diagram for explaining the transitions they have observed.
Yes, than you again for your question. This indeed looks strange at first, but the longer you live with this, the more natural it comes. We believe that this is a continuation of the previous question with the same underlying idea of relieving the stress when it becomes possible, and can suggest the following explanation. The unusually red-shifted local emission of 1 is attributed to its stressed excited state with overcrowded electron density. When DMA is moved close enough to participate in sharing electronic density, the stress is relieved as it was relieved after insertion of the inner phenylene. This interaction also gives rise to exciplex with its emission band located where it may be expected for a would-be relieved structure having a not so red-shifted local emission. This is of course only qualitative considerations based on what we get from our single-point calculations, but they seem to fit well with the full set of experimental results/calculation results. We have added the following passage to the last paragraph of Section 2.8 to reflect your question: “The unexpected positioning of the exciplex emission line to the blue of the local emission for the DMA-1* exciplex may be explained by the same underlying idea of relieving the stress when it becomes possible as follows. The unusually red-shifted local emission of 1 is attributed to its stressed excited state with overcrowded electron density. When DMA is moved close enough to participate in sharing electron density, the stress is relieved as it was relieved after insertion of the inner phenylene. This interaction also gives rise to exciplex formation with its emission band located where it may be expected for a would-be relieved structure having a not so red-shifted local emission.” We have also added a new section to the Supporting Information, with juxtaposition of calculated and experimental emission spectra for single molecule/exciplex for 1, 2 and DMA that shows the calculated transitions and explains their assignment to either local or exciplex emission, and added reference to it in the main text after Fig. 7 as “Juxtaposition of calculated and experimental emission spectra for single molecule/exciplex for 1, 2 and DMA, and assignment of transitions to local or exciplex emission, are given in Section S9 of Supporting Information.”
Your request for energy level diagram has also prompted a further hypothesis based on analysis of molecular orbitals within the Fukui approach. It is possible that there is a certain vacant orbital of acceptor (i.e., electrophilic agent), which upon interaction with a filled orbital of donor results in a shift of their energies due to overlap. This agrees with the results shown in Fig. 8, when the donor and acceptor are sufficiently close the overlap is substantial, which in the figure is revealed as a red shift of exciplex emission line and a substantial increase in oscillator strength. At larger distances the energy of the corresponding orbital is not affected. As can be seen in Fig. 8, at about 5A for 2 (and even earlier for 1) the exciplex emission wavelength is already close to local emission wavelength. It can be expected that in the absence of interaction the energy of this orbital may be much larger than LUMO and it is not populated, and thus is not revealed in the emission spectra of isolated acceptors. On the other hand, in the absence of interaction with donor its structure is such that an emissive transition from this orbital is forbidden. While in principle this is all reasonable, we did not perform the actual analysis of orbitals and did not (even try to) evaluate matrix elements of overlap between possible candidate orbitals of the donor and excited acceptor, in which several (semi-)vacant orbitals are possible both above and below LUMO. We suggest that at this moment we stick with our simple idea of stress relief that fits with all the experimental/calculation results and does not exclude this additional possibility. There seems to be quite some room for further advance in calculations here, and this is certainly one of the points to keep in mind, but we are not ready to discuss it in any substantiated way now. We suggest that no further changes be made to the text with respect to this comment, if you allow.
Reviewer 2 Report
The manuscript is devoted to the first example of exciplex formation between a series of perfluorinated para-oligophenylenes C6F5-(C6F4)n-C6F5 (n=1-3) with N,N-dimethylaniline (DMA) in degassed X-irradiated n-dodecane solutions. The ability to produce exciplexes from luminophores with arbitrarily short local excited state lifetime and no restrictions on UV-Vis spectra of the participating molecules under X-rays is demonstrated. What seems interesting is the authors' comprehensive approach to performing fluorescence studies of ion pairs. Especially I would like to emphasize the experience of the successful application of quantum-chemical calculations for predicting the possibility of exciplex formation.
However, some details of the experiment do raise questions
1. lines 128-133
Why are the original spectral data of individual components given in the article for solutions in a polar solvent, when afterwards all studies are conducted in a non-polar medium?
2. Fig. 2 and its description
Why is the same excitation wavelength chosen for compounds 2-4, which for 3 and 4 does not coincide with the position of the absorption peak? is it correct to compare quantum yields under such conditions?
The graphs in Fig. 2 are normalized and do not provide information on the emission intensity, which could be of interest.
3. Fig. 3 and its description
a) It seems very strange that systems with different concentrations of the component under study at the same concentration of the electron donor (DMA) are considered for comparison.
b) The authors have interpreted the dominant shorter-wavelength emission bands in the region of 340 nm at the spectra on Fig. 3 as local emission from the two components of the mixture. However, the position and shape of this band is more similar to the emission spectrum of DMA and therefore it is not clear why for the mixture of compound 3 with DMA its intensity is higher than for individual DMA solution.
Author Response
The manuscript is devoted to the first example of exciplex formation between a series of perfluorinated para-oligophenylenes C6F5-(C6F4)n-C6F5 (n=1-3) with N,N-dimethylaniline (DMA) in degassed X-irradiated n-dodecane solutions. The ability to produce exciplexes from luminophores with arbitrarily short local excited state lifetime and no restrictions on UV-Vis spectra of the participating molecules under X-rays is demonstrated. What seems interesting is the authors' comprehensive approach to performing fluorescence studies of ion pairs. Especially I would like to emphasize the experience of the successful application of quantum-chemical calculations for predicting the possibility of exciplex formation.
However, some details of the experiment do raise questions
- lines 128-133
Why are the original spectral data of individual components given in the article for solutions in a polar solvent, when afterwards all studies are conducted in a non-polar medium?
Yes, thank you for this question. We did spectral characterization for the compounds in two solvents, non-polar n-dodecane used for the radiation chemistry (RC) experiments afterwards and polar acetonitrile, the standard spectroscopy-grade solvent for optical studies. The full sets of experimental data (absorption, emission, excitation spectra, as well as luminescence kinetics) in the two solvents, as well as characterization of the solvents themselves, are given in the Supporting Information, and it can be readily seen that the compounds show no significant solvatochromism. The “discrepancy” you rightfully mention, and in fact almost all of the issues that you’ve raised, are due to the “hybrid” (or maybe rather “chimeric”) nature of this study, which is a cross between photochemistry and radiation chemistry. It is exactly for this reason that a dedicated section “2.2. Settings, samples, and spectra of X-ray induced luminescence” had been included in the manuscript, to stress the necessity of certain compromises. Regarding the non-polar vs polar solvent for spectral characterization, RC requires that something like n-dodecane (a rather heavy linear alkane) must be used for the experiments, but this is not available as spectroscopy-grade solvent, at purity level normally expected for fluorescence studies, and cannot be made that pure by any reasonable effort. It does pass a stringent chemical purification procedure which pushes the level of impurities down to ppm, but this still gives artifacts in classic optical fluorescence experiments. On the other hand, this is at least 1 to 2 orders of magnitude below the levels of target compounds introduces into the solution in RC experiments, which is perfectly acceptable and leads to no artifacts in RC experiments. Looking at the spectra/kinetics in acetonitrile/dodecane given in parallel in two columns in SI, it is immediately seen that they are very similar, but the dodecane ones all have either a parasitic band in the spectra, or a longer-lived parasitic emission in kinetics, all traced to the properties of n-dodecane itself as also given in SI. In addition to contaminating the spectra/kinetics, this introduces slight shifts in extracted emission lifetimes for bi-exponential fits and uncertainties in calculating quantum yields due to the need to manually suppress the parasitic band during integration, while still giving approximately the same numbers as in the normal spectra/kinetics in acetonitrile. The only information actually needed from this spectral characterization for the purpose of this work is that the emission lifetimes are too short and the spectra are too overlapping, it is certainly not a quantitative study. However, we thought that a “classic” spectroscopic characterization of this set of compounds may be useful independently of the RC experiments of his work, and opted to do it the commonly accepted way in a spectroscopy-grade solvent. This information was included in the main text, together with reference to the full set of data in the two solvents provided in SI. To address your question, we have added a further statement: “The results for the two solvents are very similar, with no noticeable solvatochromism, but the spectra/kinetics for n-dodecane are less straightforward to process due to artifacts from unconventional solvent for fluorescence measurements.”
- Fig. 2 and its description
Why is the same excitation wavelength chosen for compounds 2-4, which for 3 and 4 does not coincide with the position of the absorption peak? is it correct to compare quantum yields under such conditions?
Thank you for this question. Yes, it is correct to compare thus obtained quantum yields, and it is the common routine adopted in our group. The luminescence excitation spectra for compounds 2-4 are similar to their absorption spectra (SI). Therefore, for them light is emitted from the lowest electronic excited state, according to Kasha rule. We always excite the molecules by irradiation into the region of the lowest-energy absorption band. In this case, emission quantum yield does not depend on the exact excitation wavelength. It is not necessary to excite directly at the band maximum, it is enough to hit the region of sufficient optical density, and the amount of absorbed light is automatically accounted for in the calculation of quantum yield. The optical absorption spectra for the actual samples used to obtain the presented fluorescence spectra aiming to obtain OD of ca. 0.1 at the excitation wavelength 250/270nm are given in the SI. The excitation was performed at the wavelength at which the kinetics of emission decay were recorded (250nm for 2-4), in accordance with the availability of the pulsed excitation source (EPLED). Of the EPLEDs we had available the ones at 250nm and 270nm were suitable for 2-4 and 1, respectively. To address your question, we have added the following phrase to the footnote of Table 1: “based on availability of EPLEDs used as pulsed light source for time-resolved measurements.”
The graphs in Fig. 2 are normalized and do not provide information on the emission intensity, which could be of interest.
Yes, thank you for this question. The raw emission spectra were provided in SI, together with the optical absorption spectra from the actual samples used to get the right optical density at the excitation wavelength of about 0.1. The latter is not very critical provided it is in the acceptable range (ca. 0.05 to 0.15), and varied slightly among the as-prepared samples, which causes similar variation in absolute emission intensities. A normalization on OD would have been needed anyway to compare the emission spectra, and we chose to normalize straight through to 1. The quantitative information on the emission intensity is given by quantum yields (Table 1). To address your question, we have added the following clarification to the caption of Fig. 2: “The raw emission spectra are given in the Supporting Information.”
- Fig. 3 and its description
- a) It seems very strange that systems with different concentrations of the component under study at the same concentration of the electron donor (DMA) are considered for comparison.
Yes, this certainly does seem strange in the commonly accepted photochemistry study paradigm, thank you for raising this issue. We actually did the best possible with these compounds. As described in Section 2.2, rather high concentrations of charge acceptors are required for radiation chemistry (RC) studies, and the target concentration could be reached only for DMA and 1. The concentrations for 2-4 were limited by their solubility, as described in the text and mentioned in the caption to Fig. 3. However, this is acceptable for the purpose of the described experiments due to specifics of RC reaction cascade and the focus on the exciplex emission band arising only for pairs comprising simultaneously DMA and the target compound. The capture of the solvent radical cation by DMA is the slow process requiring the 10mM concentration, and this was the same in all experiments giving the same level of DMA radical cations available for recombination. The actual number of useful recombinations potentially leading to exciplex emission is then determined by the level of captured radical anions, in turn determined by the concentrations of 1-4, and these were indeed different, although all within the acceptable range. However, once the electron has been captured by the compound and there is a DMA radical cation available, the recombination proceeds with certainty, and this is not a chemical reaction in the usual sense, but rather a uniformly efficient conversion of the result of the previous reactions into an observable. Thus the only thing that changes with changing the concentration of 1-4 (within the acceptable limits) is the absolute amount of useful recombinations, i.e., the intensity of the exciplex emission band. But no attempt is made to quantify it, and this is hardly possible at all in RC. It is also pointless to compare the intensities of the exciplex emission band and the shorter-wavelength “local emission band”, as stressed in the text several times. So yes, this looks strange, but this is normal and acceptable for RC. We suggest to not add further comments on this in the manuscript, since we did try to maintain the balance between radiation and photochemistry in the text, mostly confining the RC specifics to a dedicated section, and discussing such details elsewhere would certainly shift this balance. Let it better stay as is, if you allow.
- b) The authors have interpreted the dominant shorter-wavelength emission bands in the region of 340 nm at the spectra on Fig. 3 as local emission from the two components of the mixture. However, the position and shape of this band is more similar to the emission spectrum of DMA and therefore it is not clear why for the mixture of compound 3 with DMA its intensity is higher than for individual DMA solution.
Yes, thank you for asking. This is again related to specifics of X-ray generated luminescence experiments that are very different from conventional photochemistry. As carefully discussed in section 2.2, this “band” at 340nm is not an emission band in the spectroscopic sense, as it is a superposition of local emissions of both components (cf. the emission spectra from Fig. 2, 1-4 emit in the same region as DMA) heavily distorted by shorter-wavelength cutoff and inner filtering. Furthermore, it is excited by a multitude of channels, including recombinations in “incomplete” pairs having only one of the partners captured, and by harvesting energy from excited solvent molecules. Although the concentration of DMA is higher and it may seem to account for a larger proportion of this band, the incomplete pairs comprising DMA are formed less efficiently due to lower (molecular) mobility of the solvent radical cation in comparison to free electron, and the compounds 2-4 also have much higher molar absorption coefficients than DMA and thus are more efficient energy harvesters, compensating for their lower concentrations. On the other hand, they have lower quantum yields than DMA. It is also not clear how the interplay of inner filtering (for e about 40000, concentration 1 mM, OD is about 10 in 2 mm path length) for the two components of the mixture would come about, but the effect is most certainly non-additive. So it is probably not fair to ascribe this band to DMA, or even mostly to DMA. It is also not that one component provides the emission, and the other one, when added, partly quenches it. It would be as reasonable to expect that since both components emit in about the same range and have channels to be excited, the emission for a mixture would by stronger than for just one component. However, if an exciplex is formed, part of this intensity is borrowed into the exciplex emission band (as part of excitation-forming recombinations are used for exciplex generation), and thus the mixtures would show a weaker “local” band than the individual components would collectively produce. Overall this comparison of different “bands” in X-ray induced luminescence spectra is a dubious business, similar to mixing different pigments in painting expecting to get the right color because “red and blue mix into purple” – but the pigments do not work additively as do point emitters in oled/TV screens, and the result may be totally unexpected, if any. As with the previous question, we suggest to not discuss this further in the manuscript, as this would raise more questions than it may hope to answer.
Round 2
Reviewer 1 Report
The authors have addressed the reviewer’s comments in the revised manuscript. The reviewer agrees with the authors that a significant stress on the structure of 1* and its relaxation in the DMA-1* exciplex will be a good model for explaining their results. The reviewer recommends the revised manuscript for publication.